# COVID-19 during Early Phase of Autologous Stem Cell Transplantation

**DOI:** 10.3390/medicina57070724

**Published:** 2021-07-17

**Authors:** Sławomir Milczarek, Bartłomiej Baumert, Anna Sobuś, Ewa Wilk-Milczarek, Krzysztof Sommerfeld, Bogumiła Osękowska, Ewa Borowiecka, Edyta Paczkowska, Aleksandra Łanocha, Wojciech Poncyliusz, Konrad Jarosz, Bogusław Machaliński

**Affiliations:** 1Department of General Pathology, Pomeranian Medical University, 70-111 Szczecin, Poland; slawek.milczarek@gmail.com (S.M.); bbaumert@pum.edu.pl (B.B.); ania.sobus@gmail.com (A.S.); edyta.paczkowska@pum.edu.pl (E.P.); 2Department of Hematology and Transplantology, Pomeranian Medical University, 71-252 Szczecin, Poland; krzysztof.sommerfeld@gmail.com (K.S.); bogumilaosekowska@gmail.com (B.O.); erbxeve@gmail.com (E.B.); aleksandra.lanocha@pum.edu.pl (A.Ł.); 3Department of General and Dental Radiology, Pomeranian Medical University, 70-111 Szczecin, Poland; ewawilkmilczarek@gmail.com; 4Department of Diagnostic Imaging and Interventional Radiology, Pomeranian Medical University, 71-252 Szczecin, Poland; wponcyl@poczta.onet.pl; 5Department of Anaesthesiology and Intensive Care, Pomeranian Medical University, 71-252 Szczecin, Poland; k.jarosz@spsk1.szn.pl

**Keywords:** COVID-19, AHSCT, lymphoblastic lymphoma, chemotherapy

## Abstract

We present one of few cases of COVID-19 occurrence during the early phase of autologous hematopoietic stem cell transplantation. We observed an interesting correlation between the patient’s rapid clinical deterioration and myeloid reconstitution that cannot be assigned to engraftment syndrome. Our report emphasizes the need to investigate whether timely steroid therapy upon neutrophil engraftment in the setting of COVID-19 could limit the extent of lung injury and prevent ARDS. Furthermore, we discuss a significant issue of possible prolonged incubation of the virus in heavily pretreated hematological patients.

## 1. Introduction

As of 20 May 2021, the COVID-19 pandemic caused by SARS-CoV-2 virus has been responsible for over 3.4 million deaths worldwide [1]. The knowledge about the pathophysiology, management and treatment of the disease is increasing systematically. The patients treated for hematological diseases are at risk of a severe course of COVID-19 to a greater extent than the general population [2]. The largest observational study, implemented by Center for International Blood and Marrow Transplant Research (CIBMTR), which included 318 hematopoietic stem cell transplantation (HSCT) recipients of autologous and allogeneic HSCT, concluded that HSCT recipients who develop COVID-19 have poor overall survival [3]. There is evidence that HSCT recipients may be at higher risk of COVID-19-associated complications due to underlying disease, immunosuppressive effect of conditioning and delayed immune reconstitution [3,4,5]. The presented case concerns a patient with lymphoblastic lymphoma who developed COVID-19 disease in the early phase of autologous HSCT.

A previously healthy 23-year-old patient diagnosed and treated for lymphoblastic lymphoma of the thoracic spine Th4 with concomitant leptomeningeal infiltration, in first complete remission, was admitted to our Bone Marrow Transplant Department (BMTD) for autologous HSCT. The nasopharyngeal swab for SARS-CoV-2 by RT-PCR performed prior to admission was negative. Patient was admitted to BMTD in good clinical condition, asymptomatic, and with unremarkable laboratory findings (Figure 1a).

The myeloablative conditioning consisted of BuCyE (busulfan 3.2 mg/kg; cyclophosphamide 50 mg/kg; etoposide 200 mg/m^2^). On the day -1, the patient presented fever, non-productive dry cough and general malaise. Oxygen saturation was normal. Physical examination revealed no evidence of lower respiratory tract involvement. Testing for viral infections was performed and SARS-CoV-2 RT-PCR came back positive. Supportive therapy was the only treatment at that point. The HSCT was performed on day “0” with a total of 12.1 × 10^6^ CD34^+^/kg bw cells infused. A gradual increase of IL-6 concentration was observed, from 12.00 pg/mL on day +1 to 488.00 pg/mL on day +6. This was accompanied by an increase of ferritin and CRP level, to 5719 µg/L and 280 mg/L, respectively. Procalcitonin levels were consistently low (Figure 2).

The therapy during the neutropenic phase of HSCT consisted of broad-spectrum antibiotics, antifungal and antiviral prophylaxis and granulocyte growth factors. Due to profound immunosuppression and hypogammaglobulinemia, two transfusions of group-compatible convalescent plasma were performed on days +3 and +6, following the data indicating amelioration of the severity of the disease [6].

On day +8, the patient’s condition deteriorated. He presented dyspnoea and hypoxemia. Chest X-ray revealed bilateral pulmonary infiltrates (Figure 1b). Chest CT scan, performed on day +9, showed bilateral, diffuse ground-glass opacities, consolidations and interlobular septal thickening (Figure 1d), which are typically present in COVID-19 [7]. The abnormalities affected approximately 60% of the lung parenchyma. Bacterial cultures, viral and fungal markers were all negative. Due to worsening of patient’s respiratory status, he was supported with supplemental oxygen via high flow nasal cannulas. Pharmacological therapy consisted of remdesivir and dexamethasone in doses corresponding with current recommendations [8,9]. Tocilizumab at the time was not recommended for use in the setting of HSCT due to its suppressive effect on neutrophils and high risk of infectious complications. Due to profound thrombocytopenia, no heparin prophylaxis was administered during this period.

Interestingly, clinical deterioration was correlated with myeloid reconstitution and neutrophil appearance in peripheral blood (Figure 2). Engraftment syndrome (ES) was excluded as no other symptoms, such as skin rash, diarrhea, weight gain, jaundice and neurological manifestations were present [10] and patient’s clinical condition deteriorated while on dexamethasone. On day +10, upon progression of respiratory failure, he was admitted to Intensive Care Unit (ICU). Mechanical ventilation was initiated and continued for 10 days, mostly in prone position. Remdesivir was administered for five days and dexamethasone was continued up to 10 days. Complete engraftment criteria were met on day +20, with PLT > 20 G/L for five consecutive days. Heparin prophylaxis was initiated. During ICU treatment, the patient was afebrile, with no signs of concomitant infection, and with negative cultures and molecular tests from bronchoalveolar lavage and blood. On day +22 he was transferred to BMTD, as he partly recovered and required only high flow supplemental oxygen. A day after, a fever appeared, and the patient was eventually diagnosed with catheter-related staphylococcal (MRCNS) sepsis and probable invasive aspergillosis. He had the catheter removed and broad-spectrum antibiotics administered. Due to positive molecular and serologic tests for Aspergillus, he received voriconazole from day +24. Because of the patient’s condition and low platelet number, bronchoscopy was not performed. Chest X-ray showed massive bilateral infiltrates typical for COVID-19 (Figure 1c). After a brief improvement in clinical condition, on day +26 he developed acute respiratory failure, unresponsive to mechanical ventilation, and ultimately sudden cardiac arrest in asystole. Due to family objection, no autopsy was performed.

## 2. Discussion

COVID-19 has a broad spectrum of clinical manifestations, from benign flu-like symptoms to acute respiratory distress syndrome (ARDS) that contributes significantly to increased mortality [11]. The severity of the disease is affected mainly by age and pre-existing comorbidities. Patients with hematologic malignancies, especially lymphomas, are believed to be at higher risk of critical course of COVID-19, ICU admissions and mechanical ventilation than non-cancer patients [3,12]. Reports indicate that the mortality rate among adult patients with hematologic malignancies with COVID-19 exceeds 30% [2]. It is believed that this is a result of underlying disease, deeply impaired immune system and conditioning toxicity [2,3]. On the other hand, some observations conclude that a group of patients might be “protected” against the severe course of the disease due to attenuated cytokine response. It is suggested that cyclophosphamide might alleviate cytokine storms and spare the critical course of COVID-19 [5,13]. The treatment in COVID-19 remains controversial and recommendations vary. The data from CIBMTR provide an overview of multiple presumed COVID-19 targeted therapies used by physicians as there are no clear recommendations for HSCT recipients, especially in early transplantation setting, mostly due to sparse data and unknown risk–benefit ratio [3,14].

## 3. Conclusions

Our report highlights some intriguing insights to debate. An interesting issue to investigate is correlation of clinical deterioration and progression of respiratory failure with neutrophil engraftment. It’s worth noting, however, that apart from neutrophils’ engraftment, the simultaneous absence of suppressor lymphocytes could also contribute to the extent of lung injury. Neutrophils are the key players in innate immunity, have a distinct array of other immune roles, such as the liberation of neutrophil extracellular traps (NETs) and cytokine production to restrict virus replication [15]. Uncontrolled NET production in COVID-19 correlates with disease severity and lung injury extension [16]. Furthermore, hyperinflammation is a trigger for thrombotic complications usually noted in COVID-19 patients [15]. The most widely accepted treatment used during cytokine storm are steroids [6]. Current recommendations for general population suggest administration of dexamethasone only in patients requiring supplemental oxygen and advise against its use in early COVID-19 phases [6]. Our report emphasizes the need to investigate whether timely steroid therapy upon myeloid reconstitution in the setting of COVID-19 could limit the extent of lung injury and prevent ARDS, especially in not-yet-oxygen-dependent patients.

Parallelly, we would like to emphasize the need to further investigate the incubation period in heavily pretreated hematological patients. Our patient was negative for SARS-CoV-2 by RT-PCR on admission and positive nine days after. The SARS-CoV-2 RT-PCR was negative in every staff member and in every other patient hospitalized at that time. We believe that the patient was infected prior to hospitalization and had a prolonged incubation of the virus as previously reported [15]. Our final suggestion is that, due to complexity of the clinical situation at the early phase of HSCT, we encourage individualized treatment of COVID-19 based on the emerging data. However, further studies are needed to address this issue.

## Figures and Tables

**Figure 1 medicina-57-00724-f001:**
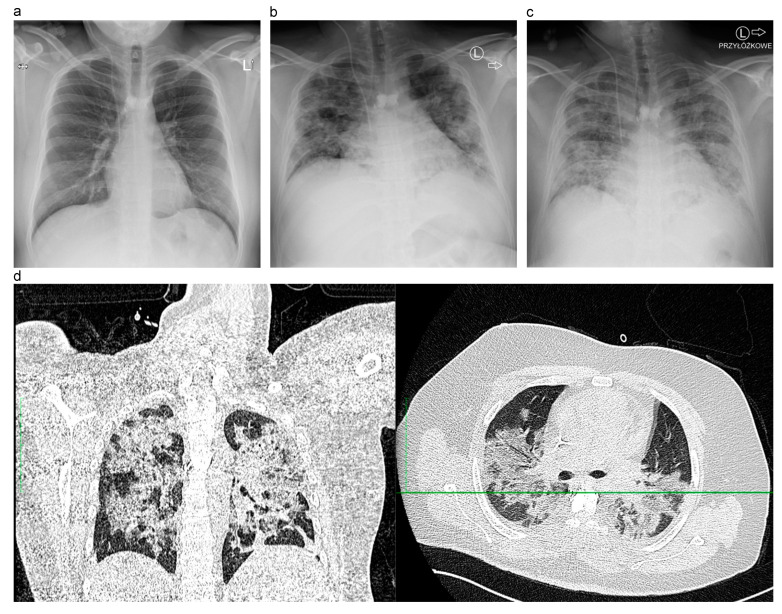
Comparison between patient’s chest X-ray imaging in different time points: (**a**) initial imaging (day -8); (**b**) clinical deterioration (day +8); (**c**) respiratory failure (day +10). Chest CT scans performed on day +9: (**d**) frontal + transverse plane. L – left site. Przyłóżkowe – bedside X-ray.

**Figure 2 medicina-57-00724-f002:**
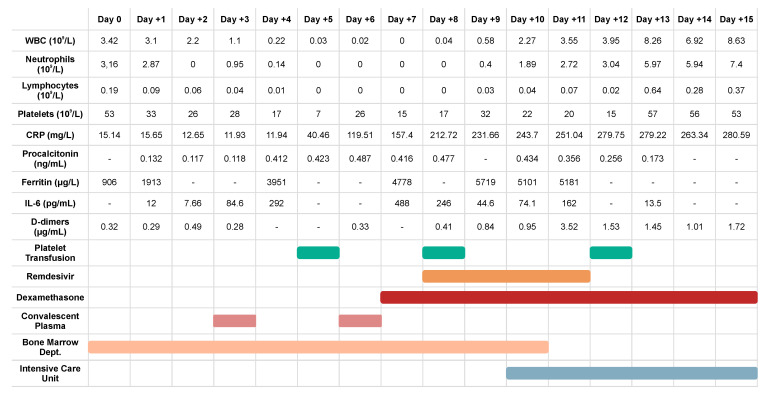
Clinical course of the HSCT (hematopoietic stem cell transplantation) recipient suffering from COVID-19.

## Data Availability

The datasets used and/or analyzed during the current study are available from the corresponding author on reasonable request.

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
