# Peer review of "COVID-19 during Early Phase of Autologous Stem Cell Transplantation"

_medicina, 2021, doi:10.3390/medicina57070724_

Round 1

Reviewer 1 Report

Covid is a new complication and the course of the disease after autologous transplant is not clearly established. In this context the submitted case report is usefull.
The case report makes the reader aware that the Covid infection is not always detected by a single screening at the moment of admission.

Author Response

Dear Reviewer,

Thank you for your comments concerning our manuscript entitled “COVID-19 during early phase of autologous stem cell transplantation”. Thank you very much for your kind opinion about our paper.

Reviewer 2 Report

The authors present an interesting report on a case of SARS-CoV-2 in a patient undergoing autologous HSCT at D-1. It is a difficult situation as standard SARS-CoV-2 therapies are tested in patients with functional immune systems, which this patient did not have early post-auto, and even once engrafted, immunosuppressive therapy would place the patient at heightened risk of infection. Therefore treatment options were limited and this report is of interest to the HSCT community.

Comments to be addressed

1 - Please include if SARS-CoV-2 test was PCR or antigen test prior to admission as these have differing sensitivity.

2 - Authors attribute the patient’s deterioration at day +8 to engraftment of neutrophils. While neutrophils are likely the cause of this deterioration, it may be that the presence of neutrophils in the absence of immunoregulatory cells that are still engrafting early post-auto exacerbated the inflammatory response in this patient.

3 – Authors suggest that use of dexamethasone upon myeloid engraftment may limit the extent of lung injury. Would this not place the patient at risk of other infections? Especially since this patient did develop sepsis at D+23.

4 - While the ARDS from SARS-CoV-2 was the likely major cause of death, what contribution did the concurrent Aspergillus infection, and treatment with voriconazole, have on patient outcome?

5 - Given the suggested longer SARS-CoV-2 incubation time in this patient, what recommendation would the authors suggest for pre-admission testing to prevent such a case from happening in the future?

Author Response

Dear Reviewer,

We would like to thank for careful and thorough reading of this manuscript and for the thoughtful comments and constructive suggestions, which help to improve the quality of this manuscript. Our response follows the comments.

Points of criticism:

  • Please include if SARS-CoV-2 test was PCR or antigen test prior to admission as these have differing sensitivity.

(The response)

We wish to thank the Reviewer for the constructive comment. According to the reviewer’s request, we have modified the Introduction and Conclusions section as follows: 

“The nasopharyngeal swab for SARS-CoV-2 by RT-PCR performed prior to admission was negative”.

“Our patient was negative for SARS-CoV-2 by RT-PCR on admission and positive 9 days after. The SARS-CoV-2 RT-PCR was negative in every staff member and in every other patient hospitalized at that time”.

  • Authors attribute the patient’s deterioration at day +8 to engraftment of neutrophils. While neutrophils are likely the cause of this deterioration, it may be that the presence of neutrophils in the absence of immunoregulatory cells that are still engrafting early post-auto exacerbated the inflammatory response in this patient.

(The response)

We wish to thank the Reviewer for the constructive comment. Indeed, this case remains unresolved as we can only hypothesize that apart from neutrophils’ engraftment, the simultaneous absence of suppressor lymphocytes can also contribute to extent of lung injury. Due to the complexity of the matter and the high likelihood that COVID cases in the early stages of AHSCT will be sporadic, it is unlikely that the phenomenon can be thoroughly investigated. However, we decided to address this issue in Conclusions section as follows:

“It’s worth noting, however, that apart from neutrophils’ engraftment, the simultaneous absence of suppressor lymphocytes could also contribute to extent of lung injury”.

  • Authors suggest that use of dexamethasone upon myeloid engraftment may limit the extent of lung injury. Would this not place the patient at risk of other infections? Especially since this patient did develop sepsis at D+23.

(The response)

We wish to thank the Reviewer for the constructive comment. Steroid therapy definitely increases the risk of infection, but since the risk of an adverse course of COVID-19 infection in stem cell recipients is high, we believe that the potential benefit may outweigh the risk. Additionally, we think that our patient died due to complications of a fungal infection he acquired in the ICU as a result of mechanical ventilation (MV), obligatory due to respiratory failure. Hence, we hypothesize that timely steroid treatment could limit the lung injury and possibly avoid mechanical ventilation.

  • While the ARDS from SARS-CoV-2 was the likely major cause of death, what contribution did the concurrent Aspergillus infection, and treatment with voriconazole, have on patient outcome?

(The response)

We wish to thank the Reviewer for the constructive comment. Unfortunately, this matter will remain unsolved as no autopsy was performed due to request of the patient’s family. As mentioned above, we believe that fungal infection and its complications played a major role in the deterioration and death of the patient. We hypothesize that pulmonary embolism may be responsible for the patient's death, as thromboembolic events typically complicate invasive aspergillosis.

  • Given the suggested longer SARS-CoV-2 incubation time in this patient, what recommendation would the authors suggest for pre-admission testing to prevent such a case from happening in the future?

(The response)

We wish to thank the Reviewer for the constructive comment. Unfortunately, it is still unclear which patients should be considered high risk for prolonged SARS-CoV-2 incubation. To prevent admissions of patients with occult infection we would suggest double RT-PCR testing in a week apart. However, the definition of high-risk group still needs to be determined.

Reviewer 3 Report

 Very interesting report.

Author Response

Dear Reviewer,

Thank you very much for your kind opinion about our paper.

We would like to thank the referees for the helpful comments and hope that our revised manuscript is now more balanced and better represents our work. We hope that the revised manuscript is acceptable for publication in Medicina.